# Cohesin cleavage by separase is enhanced by a substrate motif distinct from the cleavage site

Laura E. Rosen[1], Joseph E. Klebba[1], Jonathan B. Asfaha[1], Chloe M. Ghent[1], Melody G. Campbell [2], Yifan Cheng[2] & David O. Morgan [1]*

Chromosome segregation begins when the cysteine protease, separase, cleaves the Scc1 subunit of cohesin at the metaphase-to-anaphase transition. Separase is inhibited prior to metaphase by the tightly bound securin protein, which contains a pseudosubstrate motif that blocks the separase active site. To investigate separase substrate specificity and regulation, here we develop a system for producing recombinant, securin-free human separase. Using this enzyme, we identify an LPE motif on the Scc1 substrate that is distinct from the cleavage site and is required for rapid and specific substrate cleavage. Securin also contains a conserved LPE motif, and we provide evidence that this sequence blocks separase engagement of the Scc1 LPE motif. Our results suggest that rapid cohesin cleavage by separase requires a substrate docking interaction outside the active site. This interaction is blocked by securin, providing a second mechanism by which securin inhibits cohesin cleavage.

[1] Department of Physiology, University of California, San Francisco, CA 94143, USA. [2] Department of Biochemistry & Biophysics, University of California, San Francisco, CA 94143, USA. *email: david.morgan@ucsf.edu

The protease separase initiates chromosome segregation in anaphase by cleaving the kleisin subunit (Scc1/Rad21) of the cohesin protein complex, allowing the duplicated eukaryotic chromosomes to be segregated to opposite poles of the cell[1–3]. Tight regulation of separase function is critical, as premature cleavage of cohesin can lead to chromosome loss and genomic instability.

Separase is a large caspase-family cysteine protease (the human protein is 2120 amino acids/233 kDa). Approximately one quarter of human separase is comprised of the C-terminal protease domain, which is conserved across eukaryotes and of which it is possible to make a structural model based on homology to orthologous structures[4–7]. The large N-terminal region is poorly conserved and there is currently no detailed structural model of this region in the human protein, although it is likely composed of superhelical repeats like those seen in structures of separase from budding yeast (Saccharomyces cerevisiae)[5] and Caenorhabditis elegans[6]. Between the helical N-terminal region and the C-terminal protease domain, human separase contains regions that are predicted to be intrinsically disordered (Supplementary Fig. 1a).

Separase cleavage sites have a minimal consensus motif of ExxR, with cleavage occurring after the arginine[1,2,8]. Additional local sequence preferences have been identified[9], including an acidic or phosphorylated residue immediately upstream of the ExxR[4,8,10,11]. In the structure of the separase protease domain from the filamentous fungus Chaetomium thermophilum, basic and acidic binding pockets accommodate, respectively, the glutamate and arginine of the consensus motif[4]. Two ExxR sites are thought to be cleaved in the human Scc1 substrate[8]. Human separase contains four ExxR sites in its central disordered region, three of which are subjected to autocleavage upon separase activation[12]. After autocleavage, the N- and C-terminal domains of separase remain bound, with no apparent loss of protease activity[12]. C. elegans separase has shorter but similarly located intrinsically disordered regions, and its structure reveals that association of the N- and C-terminal domains does not depend on the disordered polypeptide chain between them[6].

In early mitosis, separase is inhibited by a high-affinity interaction with the protein securin. Securin is thought to be intrinsically disordered when free in solution[13], and the structures of securin–separase complexes from budding yeast[5] and C. elegans[6] reveal that securin binds as an extended polypeptide along the length of separase (Fig. 1a). A pseudosubstrate motif on securin interacts with the separase active site[4,14], presumably blocking substrate interactions. Securin inhibition is relieved when the N-terminal region of securin is ubiquitinated by the anaphase-promoting complex/cyclosome (APC/C) in metaphase, targeting it for destruction by the proteasome. Other vertebrate-specific modes of separase regulation have been identified, including inhibition by cyclin B-Cdk1 binding to separase in a manner dependent on proline isomerization by Pin1[15], but the specific molecular mechanism for this inhibition remains unknown.

The ExxR separase cleavage motif is ubiquitous in the proteome, but very few of these motifs are known to be cleaved by separase. Human Scc1 contains six ExxR motifs, for example, but only two are cleaved in mitosis[8]. Local sequence context at the cleavage site[9], as well as its accessibility, are expected to determine whether an ExxR motif is cleaved by separase. It also seems likely that there are other as yet unidentified mechanisms governing separase activity at the substrate level. Many proteases contain exosites: protease regions distinct from the active site that bind substrate sequences away from the cleavage site, thereby enhancing reaction efficiency[16]. The only evidence for separase regulation by substrate engagement outside of the cleavage site is that the securin-separase complex binds to DNA, helping to localize it to chromosomes[17]. While this binding results in increased cleavage of DNA-associated substrates, DNA does not enhance the enzyme's catalytic rate, and this interaction is too general to explain the observed specificity of separase.

Separase was identified two decades ago[1,2,18] and its central role in cell division is well established. However, many basic questions about its biochemical behavior and regulation remain unanswered, in part because of the difficulty of producing active protein amenable for biochemical and biophysical studies. It is well established that soluble separase can only be obtained in recombinant systems by co-expression with securin, as securin appears to be a co-translational separase-folding chaperone in addition to being an inhibitor[19,20]. Therefore, production of active separase typically begins with purification of the securin-separase complex, from which securin is removed using the APC/C-proteasome system (for human separase, an incubation with Xenopus egg extract serves this purpose)[15,21–23]. While this protocol is sufficient for certain experiments, it does not produce the quantities and purity of protein needed for detailed biophysical studies.

In the present work, we used protein engineering to develop a method for the generation of active separase starting from a set of purified proteins. Using this active separase protein we discovered that rapid cleavage of Scc1 requires a sequence motif in Scc1 that is distinct from the cleavage motif, and which we predict interacts with a docking site (exosite) on separase. We also show that securin binding interferes with separase engagement of the substrate docking motif, identifying a second mechanism by which securin inhibits cohesin cleavage by separase.

## Results

**Strategy to produce human separase for studies in vitro.** We sought to produce active human separase protein at a purity and scale sufficient for biophysical characterization. We focused on expression in Sf9 insect cells with recombinant baculoviruses[24]. First, we optimized heterologous expression of the securin-separase complex by creating a gene fusion between the securin C-terminus and the separase N-terminus, separated by a Gly-Ser linker (Fig. 1b, Supplementary Fig. 1a). This fusion was inspired by evidence that securin is a folding chaperone of separase[19,20,25] and that these protein termini are co-localized[24]. Expression of the fusion construct led to protein levels that were significantly higher than those seen when securin and separase were co-expressed in Sf9 cells (Supplementary Fig. 2). Yield was improved further by N-terminal truncation of securin to remove its APC/C degrons and by elimination of the separase autocleavage sites by mutation (Supplementary Figs. 1a, 2).

Purified securin-separase (Fig. 1c) was characterized by negative-stain electron microscopy (EM) (Fig. 1d, top, and Supplementary Fig. 3). The sample was monodisperse, and class averages were consistent with existing EM data for human securin-separase[6,24].

Human securin-separase has been demonstrated to bind DNA in a non-sequence specific manner[17]. We evaluated whether our securin-separase complex showed similar behavior. Binding of securin-separase to a fluorescently-labeled 50 base-pair double-stranded DNA molecule was evaluated by monitoring fluorescence polarization as a function of protein concentration (Fig. 1e). The data fit well to a one site specific-binding model with a $K_D$ of $300\,nM \pm 100\,nM$. A DNA molecule with the same base composition but different sequence yielded a similar $K_D$ ($220\,nM \pm 60\,nM$). Because the separase–DNA interaction is not sequence-specific, we expected that the measured affinity would depend on length, with shorter DNA molecules exhibiting lower affinities. Indeed, a 25 base-pair DNA molecule bound with a lower affinity ($K_D = 800\,nM \pm 300\,nM$).

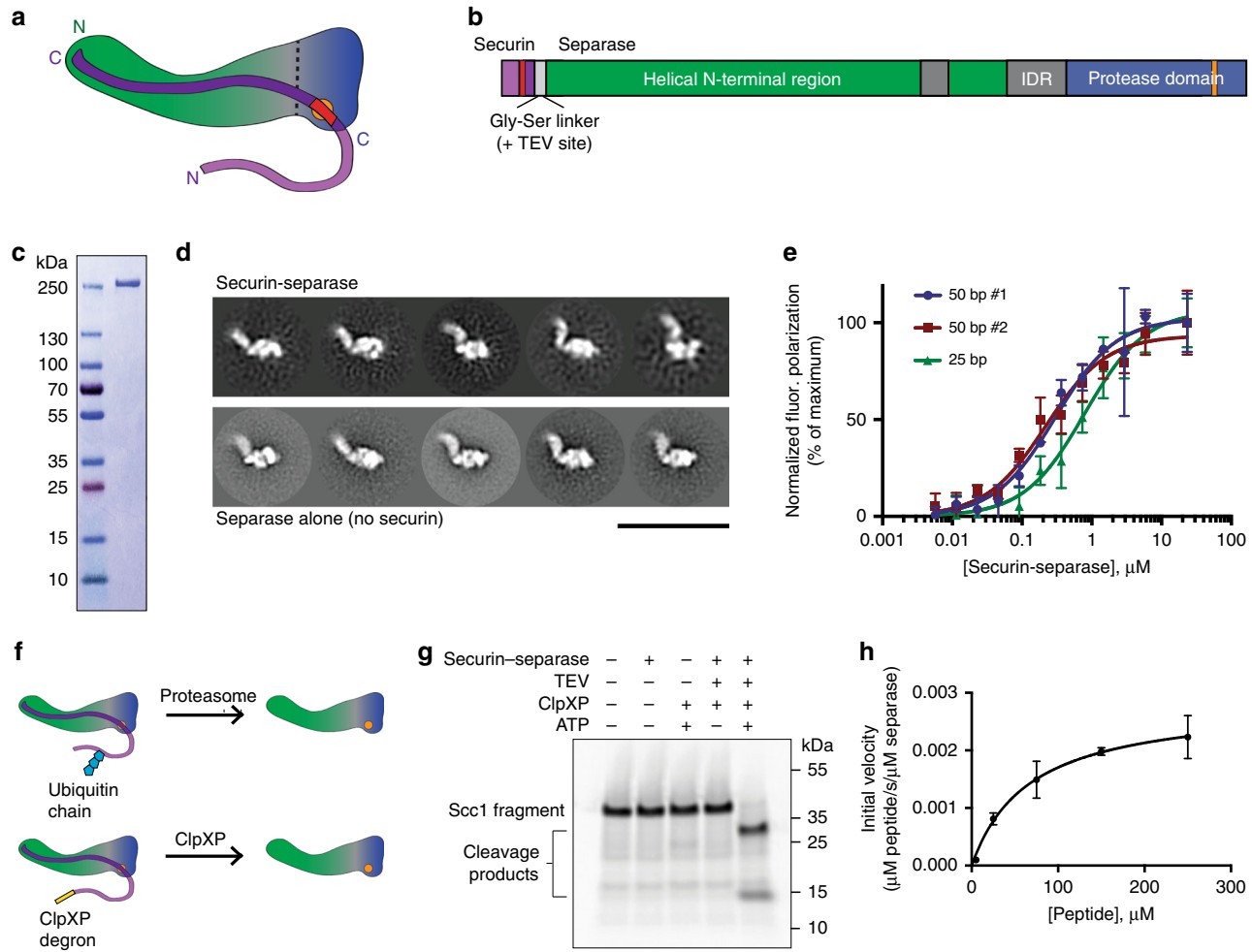

**Fig. 1** Production of active separase. **a** Cartoon of the securin-separase complex. The vertical dashed line indicates an approximate delineation between the separase N-terminal helical domain (green) and C-terminal protease domain (blue) containing the active site (orange). The C-terminal region of securin (dark purple) binds in an antiparallel fashion along the length of separase, and begins with the pseudosubstrate motif (red) bound in the separase active site. The N-terminal region of securin (light purple) contains the APC/C degrons. **b** Diagram of the securin-separase fusion construct. Colors correspond to **a**. Also depicted are the flexible Gly-Ser linker separating securin and separase (light gray) and the regions of human separase predicted to be intrinsically disordered (IDR, dark gray). See Supplementary Fig. 1a for amino acid sequence. **c** Purified securin-separase fusion protein was analyzed by SDS–PAGE and stained with Coomassie Blue, using molecular weight markers as indicated. **d** Purified securin-separase (top) and apo (active) separase (bottom) were analyzed by negative-stain EM, and five representative class averages of each preparation are shown. Scale bar: 40 nM. **e** Securin-separase binding to fluorescein-labeled DNA was evaluated by fluorescence polarization. Two 50 bp dsDNA molecules with the same base composition but different sequence were tested, as well as a 25 bp molecule. Data points indicate means (±SD) of triplicate samples. Source data are provided in the Source Data file. **f** Separase activation by the ubiquitin-proteasome system, whereby securin is tagged for degradation and removed, can be recapitulated using an N-terminal ClpXP degron and the bacterial protease ClpXP. **g** Securin-separase fusion protein was incubated with TEV protease, ATP, and/or the ClpXP ATPase as indicated, and separase activity was measured by cleavage of an $^{35}$S-labeled Scc1 fragment (residues 142–300) produced by translation in vitro. **h** Michaelis–Menten analysis was performed with purified, active separase and the peptide DDREIMREGS, which includes cleavage site 1 in Scc1. The peptide sequence was flanked by the MCA fluorophore and DNP quencher, and cleavage was monitored by an increase in fluorescence. Initial velocity was normalized to enzyme concentration. Data points indicate means (±SD) of triplicate samples. Source data are provided in the Source Data file

Next, we sought to develop a method for activating separase using purified components, rather than the traditional method of using the APC/C-proteasome system in *Xenopus* egg extract. Analogous to the proteasome, the ClpXP protein complex consists of an unfoldase (the ATPase ClpX) and a peptidase (ClpP)[26]. However, whereas the proteasome interacts with ubiquitin to determine its targets, ClpXP recognizes specific amino acid sequence motifs (degrons) on its protein targets[26] (Fig. 1f). Additionally, *E. coli* ClpXP can be produced recombinantly much more readily than the proteasome. There is also precedent for the use of ClpXP to selectively remove a protein from a protein complex[27]. We added a ClpXP degron at the N-terminus of securin in our fusion construct, as well as a TEV

protease cleavage site in the linker between securin and separase (Supplementary Fig. 1a). Following purification and cleavage with TEV protease, incubation with a purified ClpXP variant with enhanced activity towards this degron[28] removed securin and activated separase, as evaluated by cleavage of an Scc1 fragment in vitro (Fig. 1g, Supplementary Fig. 4a). Separase also cleaved a catalytically dead separase with intact autocleavage sites (Supplementary Fig. 4b), consistent with previous evidence that separase autocleavage can occur in *trans*[29].

The ClpXP-activated separase was re-purified to remove TEV protease, ClpXP, and any separase still bound by securin (Supplementary Fig. 4c-e). This purification yielded sufficient active separase to measure protein concentration spectroscopically

and to perform basic biophysical characterization. First, we used Michaelis–Menten analysis to analyze the kinetics of the interaction between the enzyme active site and a cleavage substrate. These experiments were performed with a 10-amino acid substrate peptide encompassing the best-characterized separase cleavage site in human Scc1 ([169]EIMR, or site 1) flanked by a FRET dye-quencher pair (Fig. 1h). The results fit well to a standard Michaelis–Menten curve, yielding a $K_M$ of $70 \pm 30\,\mu M$ and a $k_{cat}$ of $3 \times 10^{-3} \pm 1 \times 10^{-3}\,s^{-1}$ (or $10 \pm 3\,h^{-1}$). These results are consistent with a previous analysis of active separase reaction kinetics[30].

A 50 bp double-stranded DNA molecule did not have a significant impact on peptide cleavage reaction kinetics (Supplementary Fig. 5), consistent with previous evidence that DNA does not affect the proteolytic activity of separase[17].

Finally, we evaluated the apo separase using negative-stain EM (Fig. 1d, bottom, and Supplementary Fig. 3). The sample was monodisperse and indistinguishable from the securin-separase complex at this resolution, indicating that separase does not undergo a large-scale conformational change upon securin removal. This result seems inconsistent with the previous suggestion that securin removal causes a major conformational change[20]. It should also be noted that the separase used here is the S1126A variant, which is unable to undergo (phospho)seryl-prolyl cis-trans isomerization[15], so our results do not reflect any conformational change that may be associated with isomerization.

**Scc1 residues distant from cleavage site promote cleavage.** Our studies revealed that separase activity toward a minimal cleavage site exhibits a very low catalytic rate[31], suggesting that cleavage rate is somehow enhanced in the cell. Though it is possible that DNA binding (Fig. 1e) provides the extra affinity needed to boost function in vivo, this would be highly nonspecific if this were the only mechanism. We wanted to explore the possibility that separase has a more specific substrate docking site.

The two separase cleavage sites in Scc1 are located within a large region of predicted disorder between the terminal regions that interact with the Smc3 and Smc1 subunits of cohesin[32]. To investigate whether local sequence context accelerates the cleavage of Scc1, we evaluated a series of Scc1 truncations with an in vitro cleavage assay (Fig. 2a). Our starting point for this assay was an internal Scc1 fragment (amino acids 142–400), which was chosen after we observed more robust cleavage of Scc1 by separase when the terminal regions that interact with Smc3 and Smc1 were removed (Supplementary Fig. 4e compared to Fig. 2b). This internal fragment does not contain site 2 ([447]EPSR), and so it is cleavage at site 1 ([169]EIMR) that is being evaluated here. However, even when site 2 is present, it does not appear to be cleaved in this assay (Supplementary Fig. 4e), perhaps because cleavage at site 2 requires other factors, such as adjacent phosphorylation by Plk1[11].

We observed an abrupt reduction in cleavage of the Scc1 fragment upon C-terminal truncation from residues 275 to 250 (Fig. 2b), suggesting the presence of a separase-binding motif in this region of Scc1. Note that this assay has lower sensitivity than the above peptide cleavage assay, explaining why cleavage of the smallest fragments is not observed even though they contain the peptide sequence. Alanine scanning revealed that the most critical residues for enhanced activity were a Leu-Pro sequence at residues 255 and 256, with a contribution from Glu 257 (Fig. 2c). We will refer to these residues as the LPE motif.

**LPE motif promotes cleavage of separase biosensor in vivo.** Having demonstrated the importance of the LPE motif for separase cleavage of Scc1 in vitro, we tested its importance

in vivo. We re-created a previously described separase biosensor in human U2OS cells (Fig. 3a)[33]. With wild-type Scc1 (aa 142–467), efficient biosensor cleavage was observed during anaphase (Fig. 3b). Strikingly, the double point mutation [255]Leu-Pro→Ala-Ala reduced cleavage efficiency by 50% (Fig. 3c, d). A biosensor containing a 10 amino acid deletion centered on [255]Leu-Pro yielded identical results as the double alanine mutation, confirming that these two residues are key requirements for this interaction (Fig. 3c, d).

Although separase cleavage site 2 is present in this biosensor, it was not cleaved in our assay, nor was it cleaved in a longer version of the biosensor that extended 123 amino acids beyond site 2 (Supplementary Fig. 6). Therefore, our observation that [255]Leu-Pro promotes biosensor cleavage is specific to site 1.

**Securin inhibits separase binding to LPE motif in Scc1.** Our results suggest that an exosite on separase interacts with the LPE motif in Scc1, resulting in higher substrate affinity and more efficient cleavage. An intriguing possibility is that securin binding prevents this interaction, providing an additional mechanism by which securin inhibits Scc1 cleavage. To address this possibility, we created securin-separase fusion proteins in which securin was truncated after the pseudosubstrate sequence that binds the separase active site (Fig. 4a, Supplementary Figs. 1b, 7). The likely pseudosubstrate sequence ([113]EIEKFFP) was identified previously based on homology between human and *C. thermophilum* securins[4]. We predicted that removal of the pseudosubstrate motif would relieve the inhibition caused by securin directly blocking the separase active site, but it would retain any effect of securin binding elsewhere on separase. The fusion approach also has the benefit of generating an extremely high local concentration of securin, compared to adding securin in *trans*. We tested three securin truncations. These constructs contain securin residues 127–202, 138–202 or 160–202 covalently linked to separase via a flexible glycine-serine linker; they will be referred to as securinΔ127-separase, securinΔ138-separase, and securinΔ160-separase, respectively (Fig. 4b, Supplementary Fig. 1b).

We first asked whether removal of the securin pseudosubstrate region from the active site is sufficient to yield a cleavage-competent active site. Michaelis-Menten analyses with the peptide assay described earlier (Fig. 1g) showed that there are no significant differences between the peptide cleavage activities of the three securinΔ-separase proteins and separase with no securin bound (Fig. 4c). The securin (93–202)-separase fusion protein containing the pseudosubstrate motif was inactive in this assay (Supplementary Fig. 8). These results confirm that the pseudosubstrate sequence blocks the active site, and they also suggest that securin binding outside the active site does not impair catalysis through some allosteric mechanism, at least for the portion of securin evaluated in these experiments.

We then tested whether the securinΔ-separase constructs were able to cleave the Scc1 fragment in the gel-based assay, and whether this cleavage was sensitive to mutation of the LPE motif (Fig. 4d). SecurinΔ138-separase and securinΔ160-separase exhibited efficient cleavage of the Scc1 substrate, and in both cases activity was reduced by mutation of [255]Leu-Pro. However, securinΔ127-separase exhibited no cleavage activity in this assay. This result strongly suggests that securin interferes with separase binding to the LPE motif on Scc1, and that this interference is localized to a region of securin between residues 127 and 138. Intriguingly, this region of securin contains an LPE motif (residues 130–132).

We developed an approach to investigate the importance of [130]LPE for securin binding to separase. It is known that fungal securin can be converted to a separase substrate by making

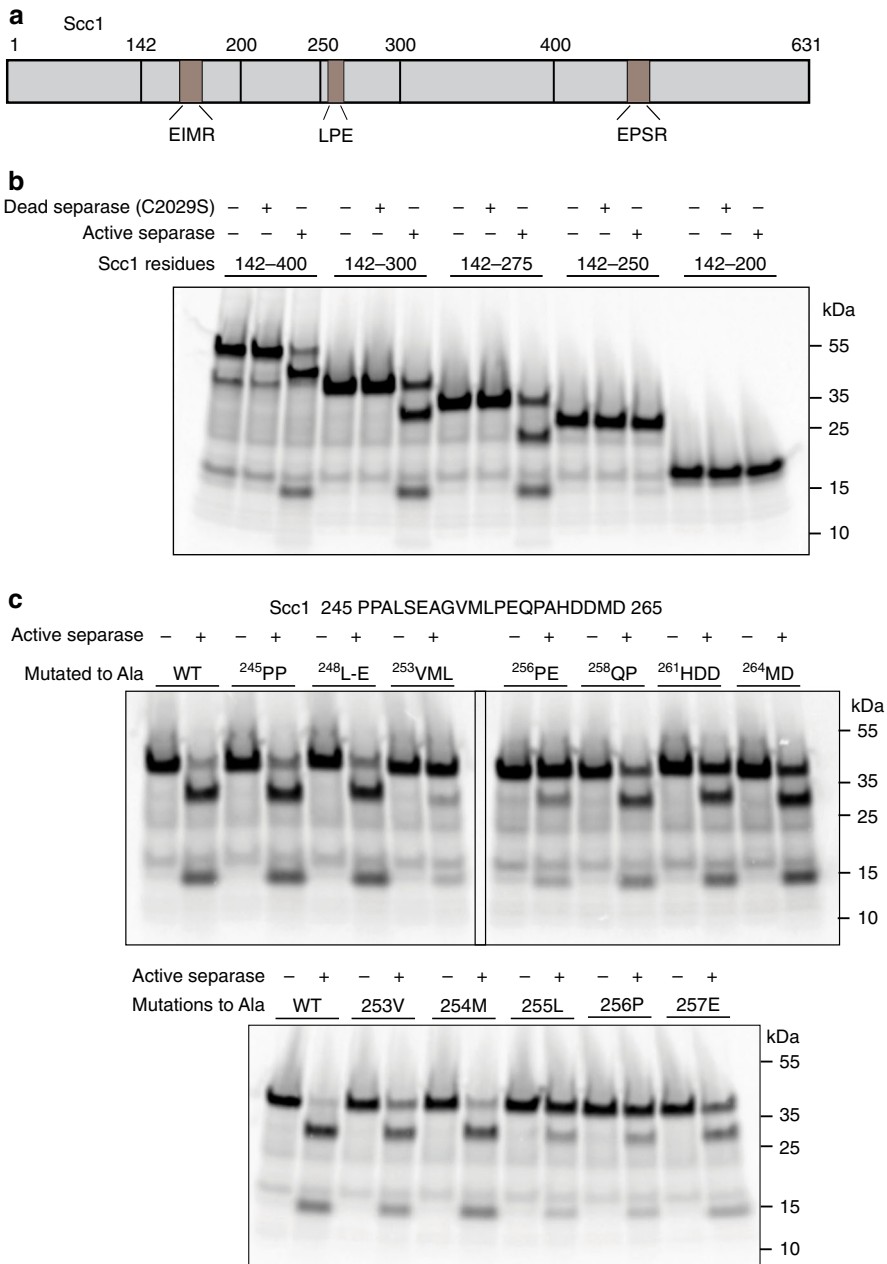

**Fig. 2** Identification of a separase docking motif on Scc1. **a** Diagram of the Scc1 sequence, showing the locations of two separase cleavage sites, LPE motif, and boundaries of truncated constructs evaluated in panels **b** and **c**. **b** [35]S-labeled Scc1 fragments were incubated with active or inactive separase as indicated, and reaction products were analyzed by SDS-PAGE and Phosphorimaging. **c** Separase was incubated with an [35]S-labeled Scc1 fragment (aa 142–300) in which the indicated residues were changed to alanines. Reaction products were analyzed by SDS-PAGE and Phosphorimaging. The sequence of the relevant region of Scc1 is shown

mutations that convert the pseudosubstrate site into a cleavage site[4,14]. We made the equivalent mutations in human securin ([118]FP to RE) and used this securin[RE] mutant to test the importance of [130]LPE for securin engagement with separase. We also tested an LP sequence a few residues further downstream ([139]LP). In pilot experiments, a securin[RE] fragment containing residues 93–202 was cleaved efficiently by separase, but mutation of either LP sequence had no effect, presumably because this fragment of securin makes too many contacts with separase for individual point mutations to significantly weaken affinity. We then tested a securin[RE] fragment containing residues 93–150. This fragment was 50% cleaved by separase, and mutation of [130]LP significantly impaired cleavage (Fig. 4e). Mutation of [139]LP

had no effect, except when combined with mutation of [130]LP. These results suggest that the [130]LPE motif of securin interacts with separase.

Consistent with its importance in the regulation of separase, the LPE sequence immediately downstream of the pseudosubstrate motif is conserved in securin from vertebrates and in some lower eukaryotes (Fig. 4f). Budding yeast securin carries a VPE sequence at this location, and the crystal structure of the yeast separase-securin complex indicates that the motif interacts with the surface of separase adjacent to the catalytic domain[5] (Supplementary Fig. 9). This region of separase is poorly conserved in human separase, precluding straightforward prediction of the LPE motif-binding site in the human protein.

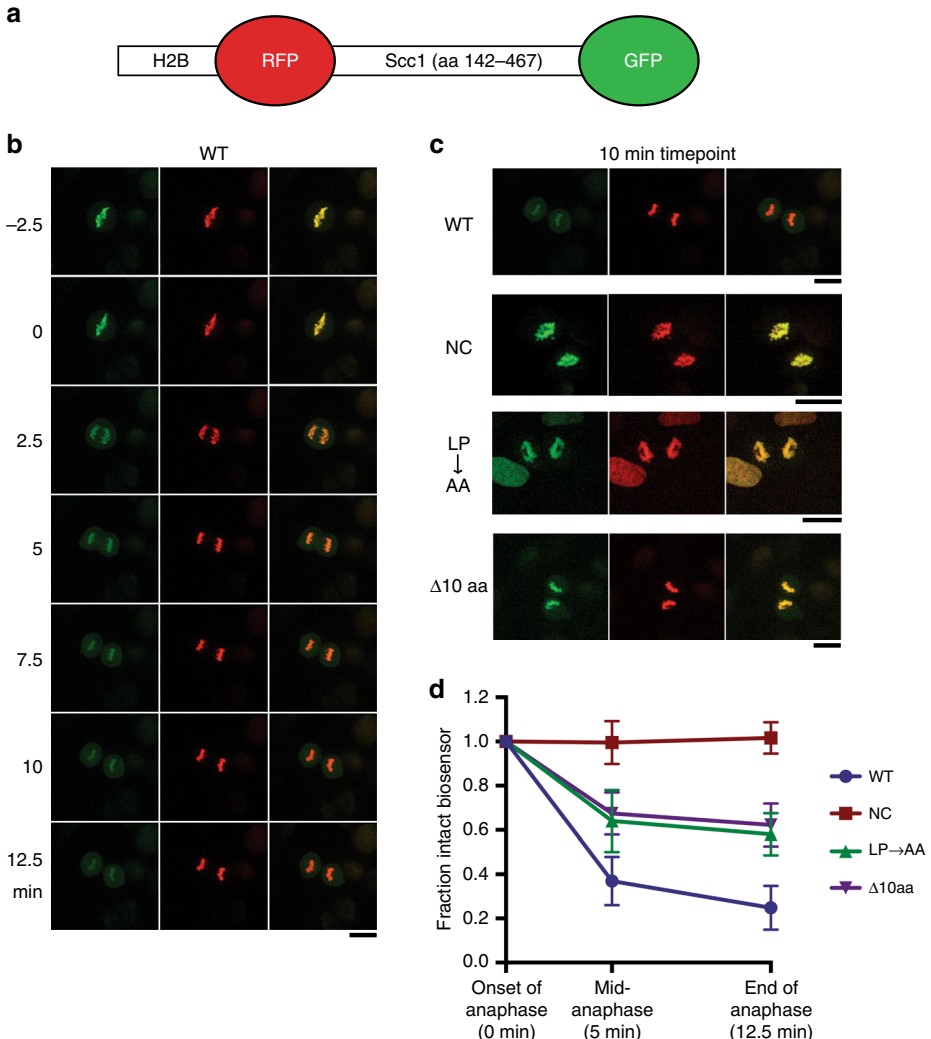

**Fig. 3** The LPE motif is important for cleavage of a separase biosensor in vivo. **a** Schematic of the separase biosensor used to evaluate cleavage in vivo, which includes histone H2B, red fluorescent protein (RFP), the indicated Scc1 fragment, and green fluorescent protein (GFP)[33]. **b** Time course of wild-type (WT) biosensor cleavage by separase, showing green fluorescence (left), red fluorescence (center), and merged images (right). Time zero is the last time point before the onset of chromosome segregation. Biosensor cleavage is indicated by reduced green fluorescence relative to red fluorescence. Scale bar: 20 μM. **c** Representative images showing late anaphase fluorescence of biosensor variants carrying mutations in Scc1 (WT, wild-type, copied from **a**; NC, non-cleavable mutations at sites 1 and 2; LP → AA, mutations of $^{255}$LP; Δ10aa, deletion of aa 251 to 260, which contain the LPE motif). Scale bars: 20 μM. **d** Quantification of the loss of GFP fluorescence in the four biosensor variants shown in **c**. Data points indicate means (±SD) from between 15 and 30 cells. Source data are provided in the Source Data file

## Discussion

Premature or partial cohesin cleavage can have disastrous consequences for a cell undergoing mitosis, and separase must therefore be tightly regulated. Several modes of separase regulation have been identified, most notably the inhibition of separase by the protein securin. Here, we identified a mechanism of separase regulation at the level of the separase substrate Scc1. Our evidence suggests that Scc1 contains a previously unknown separase-binding motif (the LPE motif), distant from the cleavage site, that enhances separase cleavage of Scc1.

Our results also clarify the mechanism by which securin inhibits separase. The C-terminal separase-interacting segment of securin binds along the length of separase in an antiparallel fashion, starting with a pseudosubstrate motif bound in the separase active site. Securin sequences downstream of this motif are extended along the surface of the N-terminal domain of separase[5–7]. It had been predicted but not previously

demonstrated that the presence of the securin pseudosubstrate motif in the separase active site was sufficient to inhibit separase catalytic activity. Our series of active separase constructs covalently bound by truncated securin (securinΔ-separase) revealed that removal of the pseudosubstrate motif alone allowed wild-type rates of peptide substrate cleavage. Thus, the remaining regions of securin do not inhibit the separase active site through an allosteric mechanism. However, in the cleavage assay with longer fragments of Scc1, these constructs showed that securin binding interferes with separase engagement of the Scc1 LPE motif, revealing a mechanism by which securin inhibits separase activity toward cohesin (Fig. 4g).

We anticipate that cleavage of other separase substrates depends on LPE or other docking motifs. In meiotic cells, the Scc1-related protein Rec8 serves as the kleisin subunit of cohesin and is cleaved by separase. The major separase cleavage site in mouse Rec8, R454[34], is followed by conserved LPE motifs about

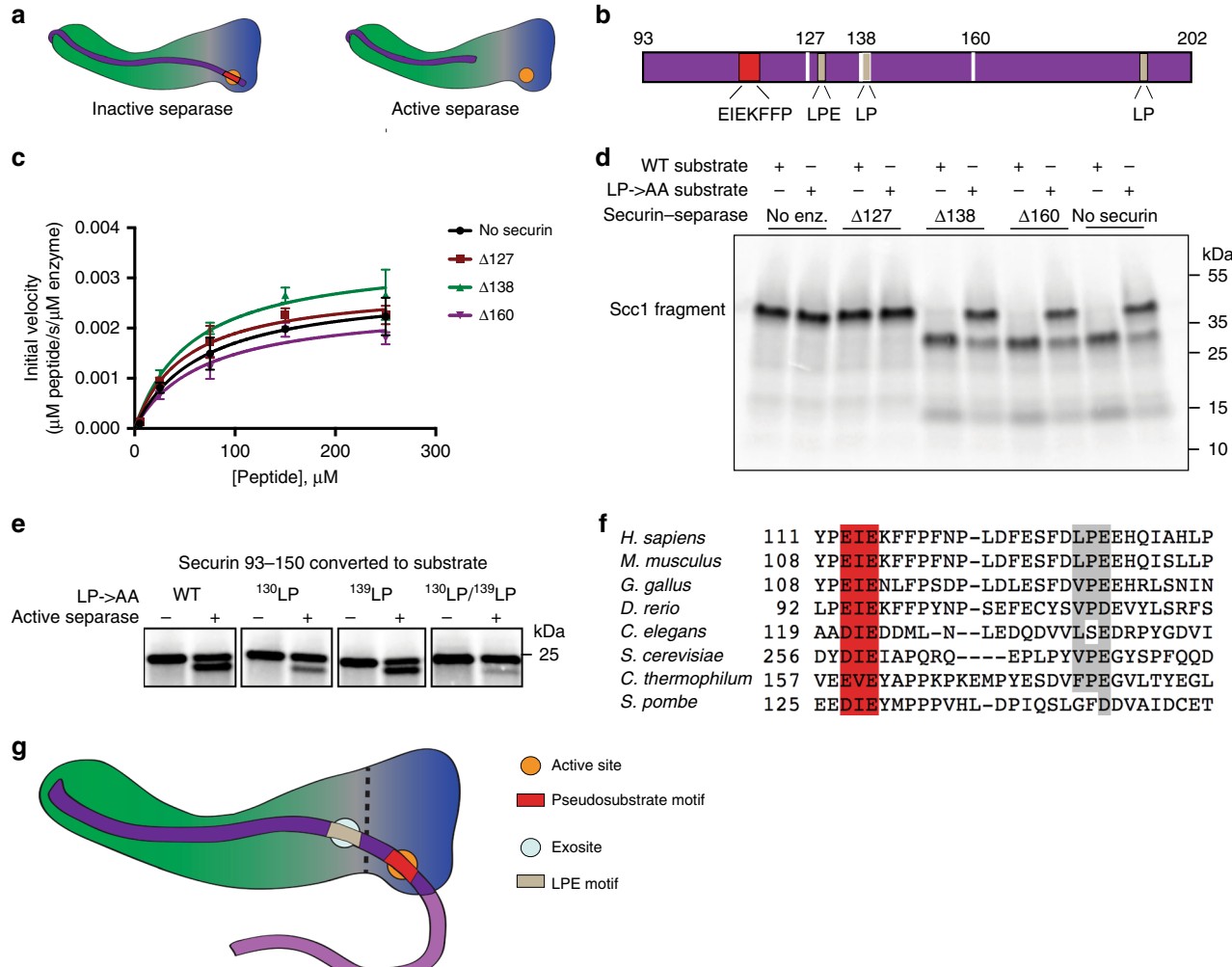

**Fig. 4** Securin inhibits binding to the LPE motif of Scc1. **a** Cartoon of the securin-separase fusion protein containing the full separase-binding region of securin (left) or with securin truncated on the C-terminal side of the pseudosubstrate motif (right). The separase active site (orange) and pseudosubstrate motif (red) are indicated. **b** Diagram of the human securin sequence, indicating the locations of the pseudosubstrate sequence (EIEKFFP), all LP sites including the $^{130}$LPE motif, and the positions of the three truncations tested. See Supplementary Figs. 1b and 7 for amino acid sequences. **c** Michaelis–Menten analysis was performed with the three indicated securinΔ-separase fusion proteins and compared with purified separase lacking securin. Initial velocity was normalized by enzyme concentration. Data points indicate means (±SD) of triplicate samples. See Supplementary Fig. 8 for a control experiment with securin that includes the pseudosubstrate motif. Source data are provided in the Source Data file. **d** $^{35}$S-labeled Scc1 fragments (aa 142–300), with or without mutations in the $^{255}$LPE motif, were incubated with the three indicated securinΔ-separase fusion proteins or with purified separase lacking securin. Reaction products were analyzed by SDS–PAGE and Phosphorimaging. **e** The pseudosubstrate motif in securin was converted to a separase cleavage site using two point mutations ($^{118}$FP to RE). Separase was incubated with an $^{35}$S-labeled securin$^{RE}$ fragment (aa 93–150) containing these mutations as well as mutations in the indicated LP motifs. Reaction products were analyzed by SDS–PAGE and Phosphorimaging. Uncropped autoradiograph is provided in the Source Data file. **f** Sequence alignment of securin pseudosubstrate motifs (red), indicating the downstream conserved LPE motifs (gray). **g** Cartoon of the securin-separase complex, illustrating the pseudosubstrate motif interaction with the active site and the LPE motif interaction with the separase exosite

50 amino acids downstream (Supplementary Fig. 10a), hinting that cleavage of Rec8 could depend on a downstream docking site like the one we describe here. It remains to be seen if LPE or other docking motifs are used to promote cleavage of cohesin across all species. Budding yeast securin contains a VPE sequence downstream of its pseudosubstrate motif (Fig. 4f, Supplementary Fig. 9), and yeast Scc1 contains an $^{376}$LPD sequence downstream of its major cleavage site ($^{265}$EQGR) (Supplementary Fig. 10b), raising the possibility that docking is a deeply conserved feature of separase function.

In vertebrate and yeast substrates, the LPE motif is located 50–100 amino acids downstream of the separase cleavage site (Supplementary Figure 10), whereas the motif is only 11–14

residues downstream of the pseudosubstrate motif in securin (Fig. 4f; Supplementary Fig. 9). Further studies will be required to assess the role, if any, of the large inter-motif region that exists in substrates but not in securin.

The LPE motif expands our understanding of the local context that allows separase to preferentially cleave one ExxR sequence over another. However, the sequence requirements for a separase substrate remain poorly understood[9]. For example, in both our in vitro and in vivo assays we observed cleavage only at site 1 even when an intact site 2 was present, and despite the fact that there are conserved LPE motifs downstream of site 2 (Supplementary Fig. 10a). There is evidence that phosphorylation near site 2 (at serine 454) is required for cleavage[11]. However, when

we moved site 2 to the local context of site 1 (i.e., changed the two residues between E and R from those present at site 1 to those present at site 2) we did not observe cleavage in our in vitro assay (unpublished results). Additionally, when we moved an extended sequence that encompasses site 1 into the context of site 2, no cleavage was observed in our in vitro assay (unpublished results). Further studies will therefore be required for a complete understanding of sequence requirements for separase cleavage.

In the course of this work, we made significant advances in the production of active separase for studies in vitro. Previous studies of active human separase required the APC/C and proteasome in a crude cytoplasmic extract for removal and degradation of securin from the securin-separase complex. Here, we were able to produce active separase using simple, purified components by employing ClpXP to remove securin. Additionally, our securinΔ-separase protein fusion constructs provide a way to easily produce constitutively active separase. We anticipate that these fusion proteins will be useful for certain types of separase studies, such as screens of small molecule separase inhibitors.

Identification of a separase docking motif on Scc1 means that separase joins the growing list of proteases that are known to employ exosite-substrate interactions distinct from the active site-cleavage site interaction[16]. This additional interaction helps explain the discrepancy between the slow separase cleavage kinetics with the peptide substrate vs. the relatively fast kinetics observed in the cell[33]. Indeed, proteases often cleave substrates in vivo at rates that are orders of magnitude faster than those observed with short peptide substrates encompassing only the cleavage site[16]. It was previously demonstrated that DNA binding helps localize separase to its substrate, which is also likely to help explain the discrepancy between rates of cleavage in vitro and in vivo[17]. The separase docking motif identified in our work provides an additional, synergistic mechanism to enhance separase engagement with its specific substrate.

## Methods

**Constructs, cloning and expression.** Securin-separase fusion constructs were cloned into a pFastBac HT A vector with an L21 leader sequence added immediately upstream of the ORF[35] (Supplementary Fig. 1a). DNA encoding the N-terminal region of each protein (containing all or a subset of the following: LambdaO ClpX sequence, 2x StrepII tag, securin, Gly-Ser linker, TEV protease cleavage site, 3x FLAG tag) was codon optimized for insect cell expression and synthesized as a gBlocks gene fragment by Integrated DNA Technologies (IDT). Separase was amplified from a human cDNA library, and mutations were made using either gBlocks gene fragments or fragment amplification and then assembled using Gibson assembly. All constructs contained the S1126A mutation to prevent (phospho)seryl-prolyl cis-trans isomerization and subsequent aggregation[15]. Catalytically-dead separase constructs contained the C2029S mutation. For all constructs with an intact active site, the autocleavage sites were mutated by reversing the E and R residues for each of the three sites[12]. All constructs were verified by full sequencing of the ~7000 bp ORFs. The resulting plasmids were transformed into DH10Bac cells to generate bacmids through in vivo recombination. Purified bacmids were used to transfect Sf9 cells and generate P1 baculovirus. For protein expression, Sf9 cells were harvested 2–3 days after infection with P2 virus. Protein expression was tested using Western blotting with anti-Strep II (StrepMAB-Classic-HRP, IBA GmbH) or anti-FLAG (Anti-FLAG M2 Monoclonal, Sigma) antibodies.

*E. coli* ClpX and ClpP-6His expression constructs were a generous gift from Andreas Martin. ClpX is the full-length, AKH version, containing the mutation R228A[28], which we modified with a C-terminal 2x StrepII tag. TEV protease construct pRK793 was a gift from David Waugh (Addgene plasmid # 8827; http://n2t.net/addgene:8827; RRID:Addgene_8827)[36]. TEV protease and ClpX were expressed in BL-21 DE3 *E. coli* at 30 °C for 4 h after induction with IPTG. ClpP was expressed in a BL21 ClpP knockout strain at 25 °C for 4 h after induction with IPTG.

The separase biosensor was generated as described by Shindo et al.[33]. Specifically, Gibson cloning was performed to generate a final construct of pCMV-H2B-mRuby2-Scc1(142–467)-mNeonGreen in a plasmid backbone containing PGK-Neo. This was used as the template for all variations of the biosensor, which were also generated using Gibson cloning.

**Protein purification.** Securin-separase fusion protein and ClpX protein were purified on a StrepTrap column, with a lysis and wash buffer of 50 mM HEPES-KOH pH 7.8, 300 mM KCl, 0.1 mM EDTA-KOH, 0.5 mM TCEP, 10% glycerol. Proteins were eluted in one step in the same buffer containing 2.5 mM desthiobiotin. Securin-separase was used for ClpXP activation (see below) or buffer exchanged via PD-10 column into relevant buffers (see below), concentrated, frozen in aliquots of 100 μl or less in liquid nitrogen (LN₂), and stored at −80 °C. Securin-separase used for negative-stain EM was additionally purified by size exclusion using a Superose 6 10/300 GL column pre-equilibrated in the following buffer: 25 mM HEPES pH 7.8, 75 mM KCl, 10 mM MgCl₂, 0.5 mM TCEP, 5% glycerol.

TEV protease and ClpP were purified on a HisTrap column. TEV protease buffers were 50 mM Tris–HCl pH 8, 200 mM NaCl, 10% glycerol, 0.5 mM TCEP, with 25 mM imidazole in the lysis and wash buffers and 800 mM imidazole in the elution buffer. ClpP buffers were 50 mM HEPES pH 7.8, 100 mM KCl, 400 mM NaCl, 10% glycerol, 0.5 mM TCEP, with 20 mM imidazole in the lysis and wash buffers and 500 mM imidazole in the elution buffer. TEV protease, ClpX and ClpP were each dialyzed overnight into 50 mM HEPES-KOH pH 7.5, 200 mM KCl, 25 mM MgCl₂, 0.1 mM EDTA, 0.5 mM TCEP, 10% glycerol. After dialysis, precipitate was pelleted by centrifugation and the supernatant frozen in aliquots of 250 μl or less in LN₂ and stored at −80 °C.

**Separase activation and purification.** Securin-separase fusion was purified as described above. Eluted fractions were stored at 4 °C overnight, and then pooled and concentrated to ~1 ml (~2.5 mg/ml). The concentrated protein was incubated with 1 ml TEV protease (~2.5 mg/ml) and 10 μl Benzonase added to 11.1 ml of 25 mM HEPES pH 7.8, 100 mM KCl, 10 mM MgCl₂, 10% glycerol for 1 h at 30 °C. ClpX (1.7 ml, ~1.6 mg/ml) and ClpP (830 μl, ~2 mg/ml) were mixed and pre-incubated at 25 °C for over 30 min. After the TEV protease incubation, 830 μl 100 mM ATP (in 25 mM HEPES pH 7.8, 100 mM KCl, 10 mM MgCl₂, 10% glycerol) was added to the securin-separase reaction mixture, followed by the pre-incubated ClpXP. After 1.5 h at 30 °C, the mixture was filtered (0.2 μm) and run on a HisTrap column to remove ClpP and TEV protease. The flow-through was pooled, concentrated to less than 2.5 ml, and run over a PD-10 column to change the buffer to 50 mM HEPES-KOH pH 7.8, 300 mM KCl, 0.1 mM EDTA-KOH, 0.5 mM TCEP, 10% glycerol. The protein was run on a StrepTrap column to remove ClpX and also any separase still bound by securin. The flow-through was pooled and concentrated to less than 1 ml, and loaded on a Superose 6 10/300 GL column pre-equilibrated in the following buffer: 25 mM HEPES pH 7.8, 75 mM KCl, 10 mM MgCl₂, 0.5 mM TCEP, 5% glycerol. The separase peak was pooled, concentrated, frozen in aliquots of 100 μl or less in LN₂ and stored at −80 °C.

**Electron microscopy.** Separase and the separase-securin complex were diluted to a nominal final concentration of 0.01 mg/ml in a buffer containing 25 mM HEPES-KOH pH 7.8, 75 mM KCl, 10 mM MgCl₂, 0.5 mM TCEP. For each sample, 3 μl was applied to carbon-coated 200-mesh copper grids (Ted Pella, Redding, California) which had been glow discharged for 30 s. Specimens were stained as previously described[37] with a solution containing 2% (w/v) uranyl formate. Data were acquired with a Tecnai F20 Twin transmission electron microscope (FEI, Hillsboro, Oregon) operating at 200 kV using SerialEM[38] and a nominal range of 0.9–1.9 μm under focus. Images were recorded on a TemCam-F816 CMOS camera (TVIPS, Gauting, Germany) at a nominal magnification of ×50,000, which corresponds to 1.57 Å/px at the detector level. For the separase sample, 337 images were collected (28,540 particles picked, ~80 particles per image) and for the separase-securin complex 75 images were collected (26,077 particles picked, ~350 particles per image). Immediately following image acquisition, micrographs were binned by two to give a final pixel size of 3.14 Å/px. The CTF was estimated using GCTF[39], and particles were picked using a reference free routine as implemented in Gautomatch (http://www.mrc-lmb.cam.ac.uk/kzhang/Gautomatch). Data were processed in a similar manner for each dataset, using Relion2[40] for 2D alignment and classification into 100 classes.

**Analysis of DNA binding by fluorescence polarization.** Double-stranded, 5′-fluorescein-labeled oligonucleotides were ordered from IDT. DNA was mixed with a dilution series of securin-separase C2029S with the following final conditions: 1 nM DNA in 25 mM HEPES pH 7.8, 50 mM KCl, 5 mM MgCl₂, 0.5 mM TCEP. Samples were incubated 30 min at 25 °C prior to measurement. Fluorescence polarization was measured on a Biotek Synergy H4 plate reader using excitation/emission of 485/528 nm at a gain of 70. Signal from wells with no protein were used to blank subtract the data, then the blank-subtracted fluorescence polarization was normalized relative to the average value at the highest protein concentration. Data were fit to a one-site binding model using GraphPad Prism.

**Scc1 cleavage assay.** ³⁵S-methionine-labeled fragments of human Scc1 (and securin; Fig. 4e) were produced in rabbit reticulocyte lysates using the TnT Quick Coupled Transcription/Translation System (Promega). Variants were made by QuikChange mutagenesis or Gibson cloning. All variants included an N-terminal ZZ tag followed by a TEV protease cleavage site. Following translation in vitro,

proteins were purified by immunoprecipitation on magnetic beads coated with anti-ZZ IgG, and eluted by TEV protease. Active separase (~0.12 mg/ml) was mixed 1:1 with purified Scc1 substrate and incubated for 1 h at 25 °C. Reaction products were analyzed by SDS-PAGE with BioRad 4–20% TXP gels and visualized with a Phosphorimager. Gels were also stained with Coomassie Blue to confirm that enzyme concentration was the same in all reactions. This experiment was also carried out in the condition where substrate was still bound to beads during separase cleavage; i.e., no TEV protease was added until after separase cleavage. The results were similar to those obtained in the presence of TEV protease.

For experiments with securin-free separase, experiments were performed either with purified active separase or with activated separase but without downstream purification to remove TEV protease and ClpXP. The presence of ClpXP had no effect on the results. Additionally, in cases where ClpXP was present, apyrase was used to remove residual ATP and thereby prevent ClpXP activity.

**Peptide cleavage assay.** The following peptide, containing Scc1 site 1, was ordered from Genscript (>90% purity): Mca-DDREIMREGS-Dnp. Peptide was dissolved in DMSO at a concentration of 47.5 mM. The peptide was serially diluted into buffer (25 mM HEPES pH 7.8, 25 mM KCl, 0.5 mM TCEP) and mixed with active separase (either securin-free separase purified after TEV protease/ClpXP incubation or purified securin∆-separase) at 0.1–0.5 mg/ml in the buffer: 25 mM HEPES pH 7.8, 75 mM KCl, 10 mM MgCl$_2$, 0.5 mM TCEP, 5% glycerol. The reaction was immediately monitored by fluorescence on a Biotek Synergy H4 plate reader, using an excitation of $328 \pm 20$ nm and an emission filter of $393 \pm 20$ nm (gain of 75). Fluorescence was monitored for 1 h with 1 min reads. Data from 5 to 30 min was used for calculation of initial velocity.

To convert relative fluorescence units (RFU) to concentration of cleaved substrate, a standard curve was generated by incubating peptide with 0.1 mg/ml Trypsin for 2 h (to achieve full substrate cleavage) and then making a dilution series (in triplicate). Fluorescence was measured on the same day and at the same gain as in the kinetic assay. A plot of RFU vs concentration of cleaved peptide was fit with a linear regression and the slope taken as the conversion factor.

Separase concentrations were measured in triplicate on a Nanodrop spectrophotometer by absorbance at 280 nm, and evaluated using a theoretical extinction coefficient at A$_{280}$ (calculated according to the number of Trp and Tyr residues)[41]. The data for the Michaelis–Menten curves were normalized by enzyme concentration. Data were fit to the Michaelis–Menten equation using GraphPad Prism. Error for reported $k_{cat}$ incorporates the error in protein concentration.

**Biosensor expression and microscopy.** Second-generation lentiviruses were generated by transient co-transfection of 293T cells (a gift of Ron Vale, UCSF) in DMEM + 10% FBS, using a three-plasmid combination: one well in a 6-well dish containing $1 \times 10^6$ 293T cells was transfected using PEI with 0.5 µg lentiviral vector, 0.5 µg psPAX and 0.5 µg pMD2.G. Supernatants were collected every 24 h between 24 and 72 h after transfection and frozen at −80 °C.

For biosensor expression, U2OS cells (a gift of Ron Vale, UCSF) growing in McCoy's media + 10% FBS were plated in a 6-well dish at $1 \times 10^6$ cells per well. The following day, 0.5 ml lentivirus was added. After 48 h incubation, media was removed and cells were washed with PBS. Next, fresh media with 500 µg/ml Geneticin was added to the cells to select for transduced cells. After 1–2 weeks of selection, cell lines were expanded for FACS analysis: cells were re-suspended in FACS sorting buffer (PBS [Ca$^{2+}$/Mg$^{2+}$-free], 1 mM EDTA, 25 mM HEPES, 1% FBS) and filtered through a 50 µM filter. These cells were then sorted on a Sony SH800 Cell Sorter, selecting for cells with moderate levels of expression.

For microscopy, U2OS cells stably expressing the biosensor were plated in 24-well glass-bottom dishes (Mattek P24G-1.0–10-F) and allowed to adhere overnight. Media was removed and the cells were washed with PBS. Media was then replaced with Opti-Mem supplemented with 10% FBS. Cells were imaged at 37 °C with 5% CO$_2$ on a Nikon Ti inverted microscope equipped with CSU-22 spinning disk confocal and EMCCD camera. Mitotic cells were identified and time points were taken every 2.5 min. For data analysis, images were processed using ImageJ software as follows. Metaphase cells were identified by visual inspection of DNA labeled with H2B-mRuby2. The mean fluorescence intensities of GFP and RFP associated with DNA was then determined and the ratio of GFP to RFP was calculated. The ratio of fluorescent intensities was normalized to metaphase ratios, as it was assumed that the biosensor was intact at this stage. For each post-metaphase time point, the GFP:RFP ratio was determined for the brightest set of chromosomes and normalized against the GFP:RFP metaphase timepoint.

**Reporting summary.** Further information on research design is available in the Nature Research Reporting Summary linked to this article.

## Data availability
The source data underlying Figs. 1e, h, 3d, 4c, e, Supplementary Figs. 2, 4d, 5, and 8 are provided as a Source Data file. All other relevant data are available within the Article and Supplementary Information files or available from the authors.

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

## Acknowledgements

We thank members of the Morgan laboratory for discussions and comments on the manuscript; Andreas Martin, Charlene Bashore, and Kris Nyquist for ClpXP reagents and advice; Sam Ivry for assistance with peptide cleavage assays; and Gira Bhabha for preliminary EM analysis. This work was supported by a postdoctoral fellowship from the American Cancer Society (to L.E.R.), an HHMI Gilliam Graduate Fellowship (to J.B.A.), and a grant from the National Institute of General Medical Sciences (R35-GM118053, to D.O.M.). Y.C. is an Investigator of the Howard Hughes Medical Institute.

## Author contributions

L.E.R. conceived the project, performed most experiments and analyzed results, with guidance from D.O.M. J.E.K. performed the biosensor experiments; J.B.A. performed the analysis of securin cleavage, C.M.G. performed some peptide cleavage assays, and M.G.C. and Y.C. performed the EM analysis. L.E.R. wrote the paper with assistance from all other authors.

## Competing interests

The authors declare no competing interests.
