## [Peer Review File · Nature Communications]

Reviewers' comments:

Reviewer #1 (Remarks to the Author):

Separase is the key cell cycle protease that triggers anaphase onset by cleaving the chromosomal cohesin complex. In this very interesting manuscript, Rosen et al. advance our understanding of human separase on a number of levels, both structurally as well as biochemically. The authors develop an ingenious method of preparing securin-free separase, something that has previously been difficult to achieve. They then use the preparation to visualize the structure of free separase using electron microscopy. For the first time, this allows a structural comparison of securin-inhibited and active separase. The authors also identify a separase substrate docking motif, aside of the cleavage site, together with its cognate docking site on separase. The experiments are performed to a high standard and support the authors' conclusions. The manuscript is well written. There is great interest in all three components of this study, the preparation of active separase, its structure and its substrate interaction. I have no hesitation to recommend publication of this manuscript in a highly visible structural and molecular biology journal, e.g. Nature Structural and Molecular Biology. Here are a few, mainly minor, comments that the authors could consider in drawing up the final version of their manuscript.

In the order of appearance rather than importance:

1. Title "Separase activity depends on a substrate docking motif...". Separase activity against a small molecule substrate would not be expected to be docking-motif-dependent. Maybe better to say that "Cohesin cleavage depends on a separase substrate docking motif...", or similar?
2. Page 2, line 2 from bottom, "An acidic or phosphorylated residue immediately upstream of the ExxR promotes cleavage..."
As well as Page 3, beginning of third paragraph, "The ExxR separase cleavage motif is ubiquitous in the proteome,..."
Both the requirement for a negative charge at -6, as well as requirements at additional positions within the cleavage motif, e.g. a hydrophobic residue at the -5 position, were first worked out by Sullivan et al. 2004, which should be cited here. There is more to the separase recognition motif than a simple ExxR. For example, the authors have themselves worked with Slk19 as a separase substrate that is cleaved at a DxxR motif. Plasticity is typical of short linear motif interactions, involving both consensus and non-consensus positions. This doesn't make substrate identification any easier, but could be discussed more broadly.
3. Page 3, second paragraph, "A pseudosubstrate motif on securin interacts with the active site,..."
". This was first shown by Nagao and Yanagida 2006, who deserve credit for their observations.
4. Page 4, line 2, "securin appears to be a co-translational separase-folding chaperone in addition to being an inhibitor". This idea was established by Hornig et al. 2002, which is relevant here.
5. Page 5, middle paragraph, "We added a ClpXP recognition site...", can the authors be specific and explain how ClpXP recognizes its substrates?
6. Figure 1c and 1f, following TEV and ClpXP treatment, the securin-separase fusion protein is expected to be shortened. I'd be curious to see what this looks like on a Coomassie gel.
7. Page 5, line 2 from bottom, "... performed with a substrate peptide...", considering what is to come, it would be interesting to state in the text how long the substrate peptide was.
8. Page 6, end of first paragraph, "... separase does not undergo a large-scale conformational change upon securin removal." This is a critically important observation. Hornig et al. 2002 concluded that a substantial conformational change takes place following securin removal. It is

worth note that this seems now rather unlikely to be the case.

9. Page 6, beginning of the new section, "Though it is possible that DNA binding provides the extra affinity needed to boost function in vivo..." It would be interesting to repeat the authors' FRET-based peptide cleavage assay in the presence of DNA. Does this alter K_M or k_{cat} ?

10. The experiments in Figure 2b, identifying the separase docking site, are performed with in vitro translated substrate fragments. It would be interesting to know more quantitatively how much the docking motif contributes to substrate cleavage. Did the authors try to purify similar substrate fragments for biochemical analysis? Any insight would be welcome. However, I appreciate the possible technical challenges and, as with point 9 above, this should be seen as optional.

11. Page 7, new section heading "The LPE motif in Scc1 is important for cleavage by separase in vivo". This is a slight overstatement; Scc1 cleavage in vivo is not really measured by this biosensor assay. This could be rephrased. In the future, it will be extremely interesting to generate a cell line in which Scc1 lacks its LPE motif.

12. Page 8/Figure 4c.

a) This is probably my most substantial concern. The experiment shown in Figure 4c is missing a positive control for separase inhibition by securin that includes the pseudosubstrate region.

b) the conclusion at the end of the paragraph reads "These results [...] suggest that securin binding outside the active site does not impair catalysis through some allosteric mechanism". It should be made clear that this is true for the portion of securin that was used in this assay.

13. Page 8, last paragraph, line 2-3, "It is known that fungal securin can be converted to a separase substrate by making mutations that convert the pseudosubstrate site into a cleavage site", this really should be followed by the Nagao and Yanagida 2006 reference.

14. a) Page 9, line 4 "...the crystal structure of the yeast separase-securin complex indicates that the valine and proline interact with the surface of separase...". It would be very interesting to see this interaction. Can the authors prepare a figure of this, based on the published structure?

b) Is a LPE or VPE sequence present in budding yeast Scc1, Rec8 or Slk19? Can the authors say anything about distance constraints between cleavage site and LPE motif?

15. Discussion, end of first paragraph, "We demonstrated the importance of this site for Scc1 cleavage both in vitro and in vivo". Compare point 11, above.

16. Discussion, middle of second paragraph, "... removal of the pseudosubstrate motif alone allowed wild-type rates of peptide substrate cleavage...". Compare point 12a), this conclusion can only be reached if a similar fusion that includes the pseudosubstrate motif indeed inhibits separase in this assay. Is this the case?

Reviewer #2 (Remarks to the Author):

Review of NCOMMS-19-14574-T by Dr Morgan and co-workers entitled "Separase activity depends on a substrate docking motif distinct from the cleavage site"

Separase, an essential Cys-endopeptidase, triggers all eukaryotic anaphases by cleaving cohesion mediating cohesin ring complexes. The few known separase substrates (Scc1/Rad21 in mitotic cohesin, Rec8 in meiotic cohesin, metazoan separase itself, pericentrin-B and CPAR-1) share an ExxR consensus site, which is necessary but by far not sufficient for cleavage. What other aspects of its substrates are recognized by the highly specific separase remains largely unknown.

To generate active, recombinant separase requires co-expression of the 'chaperone' and inhibitor securin, which then has to be removed. Until now this is achieved by incubation in anaphase extracts of *Xenopus* eggs, in which securin gets degraded via the UPS. This procedure is difficult and lengthy and yields only small amounts of active separase.

In the manuscript at hand Dr. Morgan and co-workers report an easy, extract-independent method to produce active separase. They then use this separase for biochemical characterization and electron-microscopy. They also use various truncated versions of securin and Scc1 to investigate new aspects of inhibition and requirements of substrate recognition, respectively. More specifically, they show that deletion of the pseudo-substrate sequence in securin is sufficient to enable separase to cleave a peptide substrate (but not full-length Scc1) with normal kinetics. This result demonstrates that securin binding outside the active site does not exert an allosteric inhibitory effect. Given that the catalytic dyad in the reported separase-securin structure is distorted (Luo and Tong, *Nature* 542 (2017) p255-259) this insight was not self-evident and, hence, is newsworthy. Importantly, these experiments also reveal an important LPE-motif in Scc1, which is required for efficient cleavage of (full-length) cohesin and, hence, likely entertains contacts with an exosite on separase. Finally, Dr. Morgan and co-workers unravel that securin hinders binding of separase to the LPE-motif in Scc1 and does so, most likely, by use of the same LPE-motif, which is to be found some residues downstream of the pseudo-substrate sequence.

In summary, this is a nice body of work. The method to use bacterial ClpXP to segregate and degrade securin away from separase is very elegant and will be useful for many researchers in the field. Furthermore, Dr. Morgan and co-workers reveal several new aspects of how securin inhibits separase and how separase recognizes its substrate(s).

I only have a couple of minor points, which need to be addressed prior to publication and are listed below in chronological order:

page 5/figure 1g:

It would be nice to see securin degradation, ideally a Coomassie of separase before and after treatment with ClpXP. If this is impossible for technical reasons, then can one at least demonstrate ClpXP-dependent degradation of lambdaO-tagged- but not wild type securin that was expressed in bacteria and purified?

page 5/figure S4a:

That separase can cleave itself in trans has previously been reported by Zou et al., *FEBS Letters* 528 (2002) p246-250.

page 6:

Dr. Morgan and co-workers show the first EM images of securin-free separase. Because these look very similar to those of the separase-securin complex, the authors conclude that "...separase does not undergo a large-scale conformational change upon securin removal."

While probably correct, the authors use a S1126A variant of separase, which no longer isomerizes into the short-lived, aggregation prone conformer like wild type separase does upon securin-removal. I would therefore recommend to be more careful and slightly curtail the above conclusion.

page 7/8/figure S1b:

It is not clear to me why the Tev site was moved in delta127-securin relative to the other variants.

page 11:

'proline-isomerization' should be replaced by '(phospho)seryl-prolyl cis-trans isomerization'

Figure 3:

The fluorescence images are very nice; I recommend to enlarge them relative to the cartoon of the sensor to emphasize them more.

Figure S2 is lacking a loading control

Figure S4:

A flow scheme of the subsequent purification steps (with numbers that match the lanes on the gels) would make this figure more readily accessible.

Reviewer #3 (Remarks to the Author):

Rosen et al. present a novel protein engineering strategy to produce soluble and active separase. It is a challenging task because separase needs securin as a co-translational chaperone, but securin is also a tight binder and inhibitor of separase. In previous studies, separase and securin were co-expressed and securin was degraded by the APC/C-proteasome system in crude cell lysates. The authors could strongly increase the yield of the protein with a securin-separase fusion construct lacking the ubiquitinylation site. The crude cell lysate was replaced by three purified recombinant proteins (TEV, ClpX and ClpP) to remove securin. Free separase was isolated from the uncleaved construct and from the added proteases by several chromatography steps. The resulting free separase had sufficient purity for subsequent biochemical characterization. In conclusion, this strategy has improved active separase production with regard to protein yield, purity and ease of implementation.

Minor comments:

1. Supplementary figure 4b lane 2: "Fusion protein following incubation with TEV protease." Where is the TEV-cleaved securin? The respective band should be marked or the authors should explain why it is not visible.
2. A mutant ClpX was used in this study, which has enhanced activity against proteins bearing the LambdaO degron. This should be mentioned and the exact position of the mutation (e.g. R228A) should be added.
3. Write the full names of species at first mention in the text.

Response to Reviewers

Rosen et al. NCOMMS-19-14574

We thank all three reviewers for their positive and insightful comments, which have been addressed as outlined below.

Reviewer #1(Remarks to the Author):

1. Title “Separase activity depends on a substrate docking motif...”. Separase activity against a small molecule substrate would not be expected to be docking-motif-dependent. Maybe better to say that “Cohesin cleavage depends on a separase substrate docking motif...”, or similar?

We agree that the original title was potentially misleading, and we have adjusted it as suggested.

2. Page 2, line 2 from bottom, “An acidic or phosphorylated residue immediately upstream of the ExxR promotes cleavage...”

As well as Page 3, beginning of third paragraph, “The ExxR separase cleavage motif is ubiquitous in the proteome...”

Both the requirement for a negative charge at -6, as well as requirements at additional positions within the cleavage motif, e.g. a hydrophobic residue at the -5 position, were first worked out by Sullivan et al. 2004, which should be cited here. There is more to the separase recognition motif than a simple ExxR. For example, the authors have themselves worked with Slk19 as a separase substrate that is cleaved at a DxxR motif. Plasticity is typical of short linear motif interactions, involving both consensus and non-consensus positions. This doesn't make substrate identification any easier, but could be discussed more broadly.

This is an excellent point and we have adjusted the text and cited Sullivan et al. 2004 in multiple locations in the Introduction (paragraphs 3, 5) and in the Discussion (paragraph 5).

3. Page 3, second paragraph, “A pseudosubstrate motif on securin interacts with the active site,... “. This was first shown by Nagao and Yanagida 2006, who deserve credit for their observations.

We apologize for the oversight: this paper is now cited in multiple locations.

4. Page 4, line 2, “securin appears to be a co-translational separase-folding chaperone in addition to being an inhibitor”. This idea was established by Hornig et al. 2002, which is relevant here.

This paper is now cited whenever we discuss the chaperone function of securin.

5. Page 5, middle paragraph, “We added a ClpXP recognition site...”, can the authors be specific and explain how ClpXP recognizes its substrates?

We changed the text (Results, paragraph 4) to more clearly state that ClpXP interacts with specific amino acid sequence motifs or ‘degrons’, and we now mention that we used a ClpXP variant with higher activity toward the LambdaO degron used in our work. A new figure panel (Supplementary Figure 4a) documents pilot studies in which two ClpXP degrons were tested.

6. Figure 1c and 1f, following TEV and ClpXP treatment, the securin-separase fusion protein is

expected to be shortened. I'd be curious to see what this looks like on a Coomassie gel.

The small molecular weight decrease that results from TEV cleavage can be seen in the new Supplementary Figure 4a.

7. Page 5, line 2 from bottom, "... performed with a substrate peptide...", considering what is to come, it would be interesting to state in the text how long the substrate peptide was.

Good point – we have added the length of the peptide to the text as suggested. The sequence of the peptide is provided in the Methods section.

8. Page 6, end of first paragraph, "... separase does not undergo a large-scale conformational change upon securin removal." This is a critically important observation. Hornig et al. 2002 concluded that a substantial conformational change takes place following securin removal. It is worth note that this seems now rather unlikely to be the case.

We have added a brief statement about this hypothesis, and a citation of Hornig et al., to the Results paragraph where the EM result is described.

9. Page 6, beginning of the new section, "Though it is possible that DNA binding provides the extra affinity needed to boost function in vivo..." It would be interesting to repeat the authors' FRET-based peptide cleavage assay in the presence of DNA. Does this alter K_M or k_{cat} ?

This is an excellent suggestion and we did the experiment, which is presented in a new Supplementary Figure 5. DNA had no effect on the peptide cleavage kinetics. This is consistent with the work of Sun et al. 2009, who originally demonstrated DNA binding by separase and showed that DNA does not seem to affect its catalytic activity.

10. The experiments in Figure 2b, identifying the separase docking site, are performed with in vitro translated substrate fragments. It would be interesting to know more quantitatively how much the docking motif contributes to substrate cleavage. Did the authors try to purify similar substrate fragments for biochemical analysis? Any insight would be welcome. However, I appreciate the possible technical challenges and, as with point 9 above, this should be seen as optional.

This is an excellent point and we would certainly like to have a more quantitative understanding of the effect of the LPE motif on Scc1 cleavage. We would predict, for example, that the motif enhances substrate affinity and thereby reduces K_M . To this end, we have made attempts to develop assays using bacterially expressed Scc1 fragments fused to GST or to a Halo tag. These proteins have thus far proven to be poor substrates in vitro, and the quantitation of cleavage products is far less sensitive and rigorous than the FRET-based peptide cleavage assays used in our paper. Nevertheless, we can imagine many other approaches that might be productive, and the development of quantitative Scc1 cleavage assays remains a top priority for the future.

11. Page 7, new section heading "The LPE motif in Scc1 is important for cleavage by separase in vivo". This is a slight overstatement; Scc1 cleavage in vivo is not really measured by this biosensor assay. This could be rephrased. In the future, it will be extremely interesting to generate a cell line in which Scc1 lacks its LPE motif.

Excellent point: we have rewritten the section heading to state that the motif is important for cleavage in vivo of the biosensor, not Scc1. We also agree that cell lines expressing mutant Scc1 will be critical for future studies.

12. Page 8/Figure 4c.

a) *This is probably my most substantial concern. The experiment shown in Figure 4c is missing a positive control for separase inhibition by securin that includes the pseudosubstrate region.*

Another excellent point. We did the experiment, and the results are presented in a new figure (Supplementary Fig. 8). A securin-separase fusion protein containing the pseudosubstrate motif (aa 93-202) has no detectable activity in the peptide cleavage assay.

b) *the conclusion at the end of the paragraph reads “These results [...] suggest that securin binding outside the active site does not impair catalysis through some allosteric mechanism”. It should be made clear that this is true for the portion of securin that was used in this assay.*

We adjusted the text at the end of this paragraph to state that our conclusion applies only to the region of securin analyzed in these studies.

13. Page 8, last paragraph, line 2-3, *“It is known that fungal securin can be converted to a separase substrate by making mutations that convert the pseudosubstrate site into a cleavage site”, this really should be followed by the Nagao and Yanagida 2006 reference.*

Done.

14. a) Page 9, line 4 *“...the crystal structure of the yeast separase-securin complex indicates that the valine and proline interact with the surface of separase...”. It would be very interesting to see this interaction. Can the authors prepare a figure of this, based on the published structure?*

We completely agree that this would be an interesting figure, and we have now added it (Supplementary Figure 9). The figure illustrates beautifully how the VPE motif of yeast securin fits in a complementary pocket on the surface of separase.

b) *Is a LPE or VPE sequence present in budding yeast Scc1, Rec8 or Slk19? Can the authors say anything about distance constraints between cleavage site and LPE motif?*

Supplementary Figure 10b now provides the sequence of cleavage sites and potential LPE motifs in three known yeast substrates (Scc1, Rec8, Slk19). As in the vertebrate substrates, the motif is generally about 50-100 amino acids downstream of the cleavage site – in striking contrast to the LPE motif in securin, which is quite close to the pseudosubstrate sequence. We have added a small new paragraph to the Discussion (Paragraph 4) in which we mention these distance issues in substrates and in securin. We have not yet explored the distance requirements but are planning to do so in the near future.

15. Discussion, end of first paragraph, *“We demonstrated the importance of this site for Scc1 cleavage both in vitro and in vivo”. Compare point 11, above.*

This sentence has been removed, and we no longer make any unwarranted claims about the importance of the motif for Scc1 cleavage in vivo.

16. Discussion, middle of second paragraph, "... removal of the pseudosubstrate motif alone allowed wild-type rates of peptide substrate cleavage...". Compare point 12a), this conclusion can only be reached if a similar fusion that includes the pseudosubstrate motif indeed inhibits separase in this assay. Is this the case?

As stated for point 12(a) above, we have added a new experiment that addresses this point.

Reviewer #2 (Remarks to the Author):

page 5/figure 1g:

It would be nice to see securin degradation, ideally a Coomassie of separase before and after treatment with ClpXP. If this is impossible for technical reasons, then can one at least demonstrate ClpXP-dependent degradation of lamdaO-tagged- but not wild type securin that was expressed in bacteria and purified?

A new figure (Supplementary Fig. 4a) addresses this point, at least in part. Treatment with TEV protease alone, without ClpXP, results in the appearance of a very faint truncated securin band on the gel (Lanes 2, 3). Unfortunately, the effects of ClpXP on this band cannot be assessed because the abundant ClpP protein is about the same size as securin. We attempted to use western blotting to measure securin levels before and after ClpXP, but we could not detect the protein. However, there is other evidence on this gel for the effects of ClpXP: if TEV protease is not added (lanes 4, 6), then ClpXP proteolyzes the entire securin-separase fusion protein, particularly when we used the LambdaO degron. Given this result, and given that TEV/ClpXP treatment clearly activates separase (Fig. 1g), we are confident that a significant portion of the detached securin fragment is destroyed by ClpXP.

page 5/figure S4a:

That separase can cleave itself in trans has previously been reported by Zou et al., FEBS Letters 528 (2002) p246-250.

Excellent point – we have added the citation.

page 6:

Dr. Morgan and co-workers show the first EM images of securin-free separase. Because these look very similar to those of the separase-securin complex, the authors conclude that "...separase does not undergo a large-scale conformational change upon securin removal." While probably correct, the authors use a S1126A variant of separase, which no longer isomerizes into the short-lived, aggregation prone conformer like wild type separase does upon securin-removal. I would therefore recommend to be more careful and slightly curtail the above conclusion.

This is an excellent point, and we have added a new sentence to the end of the paragraph emphasizing that our studies focus entirely on the pre-isomerization form of separase.

page 7/8/figure S1b:

It is not clear to me why the Tev site was moved in delta127-securin relative to the other variants.

The reasons for the placement of the TEV site in this construct are now explained in the legend for Supplementary Fig. 1b. In brief, we did this to allow easy removal of the N-terminal Strep tag because we worried that the tag might interfere with the separase active site. Those worries proved to be unfounded.

page 11:

'proline-isomerization' should be replaced by '(phospho)seryl-prolyl cis-trans isomerization'

Done.

Figure 3:

The fluorescence images are very nice; I recommend to enlarge them relative to the cartoon of the sensor to emphasize them more.

Done.

Figure S2 is lacking a loading control

We did not do a loading control for this experiment, but overexposure of the western blot reveals a background band that shows that roughly equal amounts of insect cell lysate were loaded in these lanes. A new panel showing this band has been added to the figure.

Figure S4:

A flow scheme of the subsequent purification steps (with numbers that match the lanes on the gels) would make this figure more readily accessible.

A flow chart has been added to Supplementary Fig. 4c.

Reviewer #3 (Remarks to the Author):

1. Supplementary figure 4b lane 2: "Fusion protein following incubation with TEV protease." Where is the TEV-cleaved securin? The respective band should be marked or the authors should explain why it is not visible.

In this figure (now Supplementary Fig. 4c), the securin band is now labeled, as in the new Supplementary Fig. 4a.

2. A mutant ClpX was used in this study, which has enhanced activity against proteins bearing the LambdaO degron. This should be mentioned and the exact position of the mutation (e.g. R228A) should be added.

This is an excellent point and we have added a mention of the mutation to the Results and Methods sections.

3. Write the full names of species at first mention in the text.

Done.

REVIEWERS' COMMENTS:

Reviewer #1 (Remarks to the Author):

I have read the revised manuscript. The authors have made an exemplary effort to address all concerns. I can strongly recommend this beautiful manuscript for publication.

Reviewer #2 (Remarks to the Author):

Morgan and colleagues have adequately addressed all points raised by this referee. Although it would still be nice to demonstrate more convincingly the ClpXP-dependent degradation of degra-tagged securin, their argumentation is valid for why this is technically challenging. Overall, the authors have further improved their already fine manuscript, so that this nice body of work is now fit for publication in Nature Communications.

Reviewer #3 (Remarks to the Author):

The revised version provides satisfying answers to all previous concerns.

Response to reviewers' comments:

The reviews were positive and no changes were requested.

REVIEWERS' COMMENTS:

Reviewer #1 (Remarks to the Author):

I have read the revised manuscript. The authors have made an exemplary effort to address all concerns. I can strongly recommend this beautiful manuscript for publication.

Reviewer #2 (Remarks to the Author):

Morgan and colleagues have adequately addressed all points raised by this referee. Although it would still be nice to demonstrate more convincingly the ClpXP-dependent degradation of degron-tagged securin, their argumentation is valid for why this is technically challenging. Overall, the authors have further improved their already fine manuscript, so that this nice body of work is now fit for publication in Nature Communications.

Reviewer #3 (Remarks to the Author):

The revised version provides satisfying answers to all previous concerns.